# Effect of Boron on the Grain Refinement and Mechanical Properties of as-Cast Mg Alloy AM50

**DOI:** 10.3390/ma12071100

**Published:** 2019-04-03

**Authors:** Shuo Zhang, Jiangfeng Song, Hongxin Liao, Yanglu Liu, Gen Zhang, Shida Ma, Aitao Tang, Andrej Atrens, Fusheng Pan

**Affiliations:** 1National Engineering Research Center for Magnesium Alloy, Chongqing University, Chongqing 400045, China; shuosurezs@163.com (S.Z.); hongxinliao@foxmail.com (H.L.); 20150902051t@cqu.edu.cn (Y.L.); gen.zhang@hzg.de (G.Z.); 13220344923@163.com (S.M.); tat@cqu.edu.cn (A.T.); 2The First Sub-Institute of Nuclear Power Institute of China, Chengdu 610213, China; 3Materials Engineering, School of Mechanical and Mining Engineering, The University of Queensland, St Lucia, Qld 4072, Australia; andrejs.atrens@uq.edu.au

**Keywords:** AM50, Boron, grain refinement, mechanical properties

## Abstract

The effect of B addition on the microstructure and mechanical properties of AM50 was investigated, and the mechanism of grain refinement was clarified. Optical microscopy, X-ray diffraction, scanning electron microscopy, and electron probe microanalysis were used to characterize the microstructure evolution. The grain size of as-cast AM50 decreased from 550 μm to 100 μm with the B content increasing from 0 to 0.15 wt.%. AlB_2_ particles in the Al-3B master alloy transformed to Mg-B, and acted as the grain refiner. The addition of B to as cast AM50 alloy results in improved mechanical properties of AM50 + xB alloys. For instance, the YTS (yield tensile strength), UTS (ultimate tensile strength), and elongation of as cast AM50 + 0.15 wt.% B alloy was 94 MPa, 215 MPa, and 12.3%.

## 1. Introduction

Mg alloys are widely used to make components in numerous industries, such as aerospace, automobile, and 3C (Computer, Communication, Consumer electronics), due to their low density, high specific strength and stiffness, good damping characteristics, electromagnetic shielding capacity, adequate formability, and good cast ability [1,2,3]. Grain refinement improves strength and ductility simultaneously [4]. Grain refinement of as-cast Mg alloy is achieved by alloying, and as a result of constitutional under-cooling. Grain refiners for Mg alloys are generally divided into two types, classified by whether or not the Mg alloy contains Zr [5,6,7]. Zr is an effective grain refiner [8] but Zr cannot be used for refining Al-containing Mg alloy [9], because Zr reacts with Al to form Al_3_Zr. An effective grain refiner of Al containing Mg alloys is needed. Furthermore, for the Al-bearing Mg alloy, there is still no commercially used grain refining technique, although several methods have been developed [10,11,12]. For instance, Zhang et al refined the grains of AZ91 with melt conditioned high pressure die casting [10], Jiang et al. used Al2Y as a grain refiner for Mg–Y–Al alloys [12].

Carbon inoculation is reported to be an effective method of refining Mg–Al alloys because of its practical advantages such as low operating temperature, large melt volume, and less fading with long holding times [13,14]. However, the addition of carbon agents such as C6Cl6 and CCl4 causes environmental problems due to chlorine emission. B-containing compounds have great potential to be used as a grain refiner for Al containing magnesium alloys, because B tends to react with Mg or Al to form diborides (MgB_2_, AlB_2_) with a close packed hexagonal structure, which is a potential heterogeneous nucleation site for the Mg alloy. Besides, the low price of B is also an advantage for the industrialization of grain refined magnesium alloys in the future. 

Several researchers have proved the grain refinement of B on as cast AZ91 alloys [15,16,17]. Suresh et al. [15] reported that a B containing master alloy such as Al-4B could refine the grains of the as cast AZ91 alloy. They found that increasing the B content from 0 to 0.04 wt.% refined the grain size from 100 μm to 30 μm; and increased the yield tensile strength, ultimate tensile strength, and the elongation from 95 MPa to 113 MPa, 118 MPa to 229 MPa, 3.3% to 4.9%, respectively. They suggested that the AlB_2_ particles were the grain refiner based on the Energy Dispersive X-Ray Spectroscopy (EDX)analysis of a particle, but more convincing proof is needed. Similar grain refinement was also obtained by Zhang et al. [16] by the addition of 0.01 wt.% B refining the grain of as cast AZ91 alloy from 120 μm to 30 μm. Zhang et al. also claimed that the grain refiner was AlB_2_ particles without any direct proof. Thus, the mechanism of grain refinement from B containing compounds is still not clear. Zhang et al. also found that a higher content of B (0.05 and 0.1 wt.%) caused grain coarsening in AZ91. 

AM50 is one of the most widely used cast magnesium alloys in the industry. However, die cast AM50 components in automotive applications suffer from casting defects like hot tearing. Grain refinement improves the strength and ductility of the alloys and also deceases hot tearing susceptibility. Therefore, this work studied the effects of B on the grain refinement of the Mg alloy AM50, and clarified the mechanism of grain refinement. 

## 2. Experimental Methods

The AM50 alloy was prepared by melting commercially available pure Mg (99.9%), Al (99.9 %), and Mg–5Mn (95 wt. % Mg and 5wt. % Mn) in a steel crucible (Machinery plant, Chongqing, China), an electrical resistance furnace (Experimental electric furnace factory, Shanghai, China) under a protecting gas, which consisted of 99% CO_2_ and 1% SF_6_. The resistance furnace was heated to 670 °C, and pure Mg was put into the crucible to be melted, then Al and Mg–5Mn was added to the Mg melt and then different amounts of the Al–3B master alloy was added into the melt at a temperature of around 730 °C. The melt was continuously stirred for 10 min, held for 15 min, and poured into a steel cylindrical mold, which was preheated to 300 °C. The casting was cooled in air and removed from the mold. The result of the actual chemical composition of AM50 + xB% (x = 0, 0.05, 0.10, 0.15) are listed in Table 1, which were measured by XRF (X-ray fluorescence) (Shimadzu Corporation, Kyoto, Japan).

All the test samples were cut from a half radius of the ingots with a size of 10 mm × 10 mm × 10 mm. Microstructures were revealed using optical microscopy (OM) with a Zeiss Axio Observer A1 (Carl Zeiss AG, Jena, Germany) after polishing and etching with an acetic picral solution containing 3 mL acetic acid, 3 g picric acid, 45 mL ethanol, and 5 mL distilled water. The grain size was calculated by the intercept length method (IL, each grain size was calculated from around 100 counts), which measures the distance between two adjacent grain boundary intersection points on a test line segment. The grain morphologies were observed using scanning electron microscopy (SEM) using a Tescan Vega II (Tescan Orsay Holding, Brno, Czech Republic) and a JEOL JSM-700F (JEOL, Tokyo, Japan) after polishing and etching the sample with 4% nitric acid in alcohol. The phases of the experimental samples were identified using Rigaku D/max 2500 X-ray diffraction (XRD) (Rigaku Corporation, Tokyo, Japan) of the transverse section, using Cu Kα radiation (wavelength λ = 0.15406 nm) at 60 kV and 150 mA with a sample tilt angle ranging from 20° to 100°. The phase composition was characterized using an energy-dispersive spectrometer (EDS), and a JXA-8230 electron probe micro-analyzer (EPMA) (JEOL Ltd., Tokyo, Japan) with INCA and AZTEC computer programs.

Standard tensile specimens were machined with a Φ 5 ± 0.03 mm gage diameter and a 63.66 mm gage length. Tensile tests were carried out on a universal tension testing machine (SANSI UTM5000) (SUNS, Shenzhen, China) at an initial tensile strain rate of 1 × 10^−2^ s^−1^ at room temperature. At least three samples were tested for each alloy.

## 3. Results

### 3.1. Characterization of the Al–3B Master Alloy

XRD and EPMA analyses were performed to understand the microstructure and distribution of the B containing compounds in the Al–3B master alloy. Figure 1 presents the XRD results, it was found that Al (PDF# 04-0787) and AlB_2_ (PDF# 39-1483) were the main two phases. These two phases were also identified in the typical SEM micrograph, as shown in Figure 2. Figure 2a show a blocky particle, with a size of ~10 μm, identified as AlB_2_. A similar shape of a typical particle in the Al–4B master alloy was also identified as AlB_2_ by Suresh et al. [15]. The back scattered electron (BSE) mode in Figure 2b gives a contrast better than in Figure 2a between Al and AlB_2_, because AlB_2_ has a lower average relative atomic number than Al. The particle was identified as AlB_2_ by EPMA from the point analysis presented in Figure 2c,d of the particle marked with a red cross in Figure 2a. The low atomic number makes it hard to accurately detect the element B, however EMPA provides more reliable results than EDX. The point analysis shows the presence of B allowing identification of the particle in Figure 2a as AlB_2_. Thus, it was identified that the Al–3B master alloy is composed of Al and AlB_2_ particles.

### 3.2. Microstructures of the AM50 + xB Alloys

Figure 3 presents typical optical microstructures of the AM50 alloys with different amounts of B. The grain size measured by the linear intercept method decreased from 550 μm to 100 μm with increasing content of B from 0 to 0.15 wt.%. The grain sizes are summarized in Table 2. A small amount of B addition resulted in an effective grain refinement of as cast AM50. Besides, with the increase of B content, the microstructures of AM50 + xB alloys changed from coarsen equiaxed grains to granular grains. 

Figure 4 presents the stress-strain curves of the as-cast AM50 alloy with and without B additions. The YTS increased from 48 ± 5 MPa to 70 ± 3 MPa, 87 ± 2 MPa and 94 ± 2 MPa with alloying with 0.05, 0.10, and 0.15 wt.% B. Simultaneously, the ultimate tensile strength (UTS) and elongation also increased. With alloying of 0.15wt. % B, the UTS increased 37% from 157 ± 4 to 215 ± 6 MPa. The ductility also increased 33% from 8.3% to 12.3%. The tensile properties of AM50 + xB alloys are also summarized in Table 2.

### 3.3. Identification of the Grain Refiner

In order to identify the grain refiner of the effective grain refinement of B on AM50 alloys, phase composition of AM50 alloys was studied. Figure 5 presents the equilibrium phase diagram of the Mg–0.5Mn–xAl (x = 0–10) system calculated using the Pandat software. B was not included due to the lack of B data. The calculated section indicates that the AM50 alloy consists of α–Mg, Mg_17_Al_12_, and Al_4_Mn at room temperature, as indicated by the vertical arrow in Figure 5. The microstructures of AM50 + xB alloys indicate that the main phases are indeed α–Mg, Mg_17_Al_12_, and Al_4_Mn, as presented in Figure 6. This indicates that the B addition did not significantly change the phases. Nevertheless, there might be some B-containing phases in the AM50 + xB alloys.

Thus, further analysis focused on the B containing special particles located approximately in the center of the grain, which may act as a grain refiner. To assure the accuracy of B detection, the EPMA technique was utilized. One B containing particle was found in the specimen of the AM50 alloy refined by 0.15 wt.% B, as shown in Figure 7. The grain is marked with a black dashed line. The compound particle was located nearly in the center of the primary α–Mg grain, and was designated as A in Figure 7a. The magnified images are presented in Figure 7b. The EMPA line scan results for the compound particle A is presented in Figure 7c–f. The bright white particle was rich in Al and Mn, and was identified as the Al_4_Mn phase. The dark grey particle located besides the Al_4_Mn phase was rich in Mg and B and was marked with a red dashed line in Figure 7b,c.

A similar compound particle was also found in AM50 + 0.1 wt. % B alloy. Figure 8a,b shows the SEM images in SE and BSE mode, which reveals two adjacent particles in the middle of the grain, marked with a black dashed line. The details of the particle are shown in Figure 8c,d, their point EMPA analysis is shown in Figure 8e,f, and the data of point analyses are listed in Table 3. The white particles are identified as the Al_4_Mn phase and the dark gray particles near the Al_4_Mn is a B containing particle. 

Both the line scan result for the AM50 + 0.15 wt.%B alloy in Figure 7 and the point analysis in Figure 8 for the AM50 + 0.1 wt. %B alloy indicate that the dark gray particle is rich in Mg and B but not rich in Al. Since both particles were located approximately in the center of the grain and the grain size was largely refined with the addition of B, the B containing compound was considered as the grain refiner. Many previous studies indicated that the grain refiner in the Al containing magnesium alloys with B addition was AlB_2_, but the present study disagrees with their findings. The present study indicated that the grain refining particle was rich in Mg instead of Al. Both AlB_2_ and MgB_2_ have a close packed hexagonal structure, and are a potential heterogeneous nucleation site for the Mg alloy. Thus, the grain refiner in AM50 alloy in the present study was identified to be MgB_2_. 

There are two noteworthy features of the B containing compound in the AM50 + xB alloy. Firstly, the B containing compound was located next to the Al_4_Mn phase, which is probably attributed to the solidification path of the compound. Secondly, the amount of the B containing compound (Mg–B or Al–B) increased with an increase in the B content to some extent. As AM50 + 0.05 wt.% B alloy contained the smallest amount of B, such a compound was not found in the present study. The high amount of B containing compound resulted in a finer grain size, since the AM50 + 0.15 wt.%B alloy exhibited the finest grains among these alloys.

## 4. Discussion

### 4.1. Mechanism of Grain Refinement

#### 4.1.1. Formation Potential of MgB_2_


The grain refining particle was rich in Mg and B, as shown in Figure 7 and Figure 8, and the particle is most likely to be MgB_2_. Hence, it is important to understand the formation of MgB_2_ in the melt and how these particles act as a grain refiner. In order to explain the formation of MgB_2_, the thermodynamic calculation analysis was carried out. The enthalpy of the three possible B containing compounds Al–B, Mg–B, and Mn–B compounds in the AM 50 + xB alloys was calculated and compared. Some semi-empirical models for thermodynamic calculation have been developed to predict the enthalpy of alloys [18,19,20]. From these, the Miedema model [21] was selected due to its high accuracy to calculate the enthalpy of the three metal diborides in this study. 

The Miedema model indicates that the enthalpy is given by:(1)$$△H_{ij}=f_{ij}\frac{x_{i}\left[1+\mu_{i}x_{j}\left(\varphi_{i}-\varphi_{j}\right)\right]x_{j}\left[1+\mu_{j}x_{i}\left(\varphi_{j}-\varphi_{i}\right)\right]}{x_{i}V_{i}{}^{\frac{2}{3}}\left[1+\mu_{i}x_{j}\left(\varphi_{i}-\varphi_{j}\right)\right]+x_{j}V_{j}{}^{\frac{2}{3}}\left[1+\mu_{j}x_{i}\left(\varphi_{j}-\varphi_{i}\right)\right]}$$
where $x_{i}$ and $x_{j}$ are the mole fraction of the component i and j, $\mu_{i}$ and $\mu_{j}$ are the empirical constant of the valence state of the elements, $\varphi_{i}$ and $\varphi_{j}$ are the electronegativity of the component i and j, the $V_{i}$ and $V_{j}$ are the atomic volumes of the pure component i and j in the alloy system, respectively. 

The $f_{ij}$ indicates that the atom of the component i is encircled by the component j of the component, which is expressed as follow: (2)$$f_{ij}=2pV_{i}{}^{\frac{2}{3}}V_{j}{}^{\frac{2}{3}}\frac{\left\{\frac{q}{p}{\left[{\left(n_{ws}^{-\frac{1}{3}}\right)}_{j}-{\left(n_{ws}^{-\frac{1}{3}}\right)}_{i}\right]}^{2}-{\left(\varphi_{i}-\varphi_{j}\right)}^{2}-\alpha \frac{R}{P}\right\}}{{\left(n_{ws}^{-\frac{1}{3}}\right)}_{j}+{\left(n_{ws}^{-\frac{1}{3}}\right)}_{i}}$$
where p depends on the alloy formed between different elements, $n_{ws}^{\frac{1}{3}}$ is the electron density on the boundary of the atomic cell, α is an empirical constant of 0.73, $\frac{R}{P}$ is a modified parameter composed of two element alloys for non-transition and transition elements, the $\frac{q}{p}$ is an empirical constant of 9.4, and the other parameters are given in Table 4 and Table 5. All the parameters are from Reference [21]:

Figure 9 presents the calculated enthalpies, using the above equation and the listed parameters. Figure 9 indicates that the enthalpy of Mg–B is lower than that of Al–B and much lower than that of Mn–B. In other words, Mg–B is more stable than Al–B. 

Another study also showed that AlB_2_ at a temperature above 600 °C becomes less stable than at room temperature [22]. As a result, in the cast condition of this study, MgB_2_ is more likely to be present than AlB_2_. In addition, MgB_2_ can form at a low temperature of ~600 °C [23]. Furthermore, the phase transformation of MnB_2_ occurs at ~1400 °C [24], which is unlikely to occur in this work, since the cast temperature is around 700 °C. Consequently, the MgB_2_ phase is most likely to form in the current cast condition. Consequently, the AlB_2_ particle in the Al–3B master alloy reacts with Mg to form the MgB_2_ particle. That indicates that the reaction Mg + AlB_2_→ MgB_2_ + Al occurs with high tendency. 

#### 4.1.2. Grain Refining Potential of MgB_2_ and AlB_2_

Section 4.1.1 indicated that MgB_2_ can form in the casting conditions studied herein. It is also of significance to prove that MgB_2_ can act as a grain refiner. AlB_2_ has a hexagonal closed-packed structure [25,26] and belongs to the P6/mmm (NO.191) space group, which has a similar lattice structure to Mg (P63/mmc (NO.194)) [27]. Furthermore, the AlB_2_ particle is stable [15,28], which is another requirement of a grain refiner. MgB_2_ also has a hexagonal closed-packed structure, and belongs to the same space group (P6/mmm (NO.191)) as AlB_2_. It is also reported that Mg_0.5_Al_0.5_B_2_ exists as a transition phase during the phase transformation from AlB_2_ to MgB_2_ in the Mg-rich area of the phase diagram [29]. Table 6 compares the crystallographic data of AlB_2_, MgB_2_, and Mg. The misfit on (0001) of AlB_2_ and MgB_2_ with Mg is 6.36 and 3.86, respectively. According to the Bramfitt theory [30,31], a misfit less than 6% is a perfect heterogeneous nucleating agent, and a misfit between 6% and 12% is a moderately effective heterogeneous nucleating agent. As a result, MgB_2_ with a lower misfit is a more effective heterogeneous nucleating agent than AlB_2_ in the Mg melt. 

#### 4.1.3. Solidification Process and Grain Refinement

Grain refinement occurs during solidification. Thus, it is essential to clarify the solidification processes in the alloys with and without B. The B containing compound tends to appear near the Al–Mn phase, which may result from the solidification process.

In order to clarify the solidification process of AM 50 containing B, it is necessary to understand the solidification of AM50. Figure 10 shows the phase formation during the solidification of AM50 calculated using Pandat. Firstly, Al_8_Mn_5_ and α–Mg are precipitated from the liquid phase; then, with the decrease of temperature, a series of solid phase transitions occur in Al_8_Mn_5_; finally Al_4_Mn+Al_12_Mg_17_ is formed. The detailed solidification process and phase transformation is as follow: 

Solidification process: L→L+Al_8_Mn_5_→L+Al_8_Mn_5_+α–Mg→Al_8_Mn_5_+α–Mg

Phase transformation: Al_8_Mn_5_+α–Mg→Al_8_Mn_5_+α–Mg+Al_11_Mn_4_→α–Mg+Al_11_Mn_4_→α–Mg+Al_11_Mn_4_ +Al_4_Mn→α–Mg+Al_4_Mn→α–Mg+Al_4_Mn+Al_12_Mg_17_

For AM50 + xB, it is not possible to calculate the solidification process with the Pandat software due to the lack of B data in the database. However, it can be predicted. According to the previous analysis, the minor addition of B does not significantly change the microstructure, only the MgB_2_ particles are formed and act as the grain refiner. It is predicted that the AlB_2_ particles in the Al–3B master alloy react with Mg in the melt to form MgB_2_ at ~650 °C. With decreasing temperature, the Al–Mn phase tends to appear and adhere to the previously formed MgB_2_ particle. With further decreasing temperature, the phase transformation between the Al–Mn compounds occur and the MgB_2_ particle act as heterogeneous nucleation site of the primary Mg grains. 

Thus, the solidification process of AM50 + xB is predicted as follows:

L+AlB_2_→L+MgB_2_→L+MgB_2_+Al_8_Mn_5_→L+MgB_2_+Al_8_Mn_5_+α–Mg→MgB_2_+Al_8_Mn_5_+α–Mg→MgB_2_+Al_8_Mn_5_+α–Mg+Al_11_Mn_4_→MgB_2_+α–Mg+Al_11_Mn_4_→MgB_2_+α–Mg+Al_11_Mn_4_+Al_4_Mn→MgB_2_+α–Mg+Al_4_Mn→MgB_2_+α–Mg+Al_4_Mn+Al_12_Mg_17_

### 4.2. Tensile Properties

Fine grain strengthening is considered an important factor to improve the mechanical properties of as-cast Mg alloys. The fine grain strengthening is characterized by the Hall–Patch relationship [32]. The enhancement of the yield strength by decreasing grain size, *d*, is expressed as follows:(3)$$\sigma_{0.2}=\sigma_{0}+kd^{-\frac{1}{2}}\;$$
where $\sigma_{0}$ is the yield strength of the Mg alloy and *k* is the coefficient of the Hall–Petch equation (*k* = 280–320 MPa μm^1/2^ for Mg alloys) [33,34]. As a comparison, the *k* value of Al is only ~68 MPa μm^1/2^ thus that the grain strengthening of Mg alloys is more effective than that of Al alloys. Therefore, it is useful to a find effective grain refiner for magnesium alloys to improve their mechanical performance.

The tensile strength (157 MPa) and elongation (8.3%) of the as-cast AM50 was comparable to the data in the literature. Luo and Sachdev [35] measured an UTS of 170 MPa and an elongation of ~6% for as-cast AM50, whereas Mert et al. [36] measured a UTS of 145 MPa and an elongation of 2.8%. In addition, in this study, alloying with only 0.15 wt. % B, increased the tensile strength to 215 MPa and the ductility increased to 12.3%. These results indicated that the minor addition of B improved both strength and ductility due to the grain refinement. Thus, B is an effective grain refiner of as-cast AM50 alloy.

## 5. Conclusions

The influence of B addition on the microstructure and mechanical properties of as-cast AM50 alloy was revealed in the present work. The major conclusions are as follows:

(i) The microstructure of as-cast AM50 alloy mainly consists of Mg_17_A1_12_, Al_4_Mn, and α–Mg phases. The microstructure did not change significantly with the increasing B content.

(ii) Boron is an effective and cheap grain refiner for Al containing magnesium alloys, which is very important for the die cast AM50 magnesium alloy parts. Alloying AM50 with 0.15 wt.% B using the Al–3B master alloy refined the grain size from 550 μm to 100 μm; and the ultimate tensile strength and the elongation increased by 37% and 33%.

(iii) The grain refiner was identified as MgB_2_ particles instead of AlB_2_, which were located near the Al–Mn phase in the center of the grain. However, the MgB_2_ particle is not yet identified by TEM in this study, which will be further investigated in the future. 

## Figures and Tables

**Figure 1 materials-12-01100-f001:**
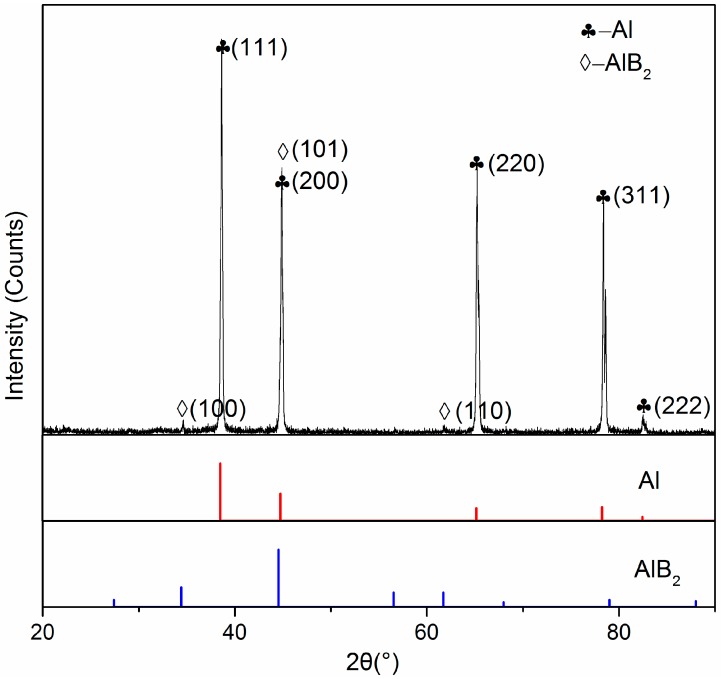
XRD results for the Al-3B mater alloy.

**Figure 2 materials-12-01100-f002:**
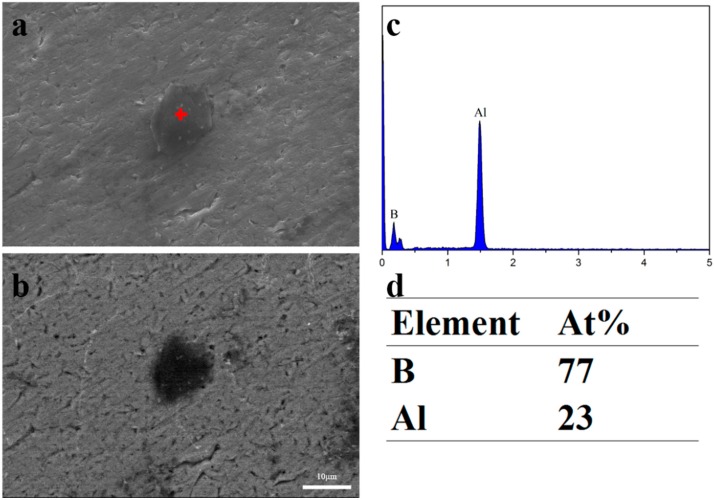
A typical SEM micrograph of the Al-3B master alloy (**a**) in SE (secondary electrons mode), (**b**) in BSE (back secondary electrons) mode, (**c**–**d**) the EPMA point analysis of the particle marked in (a).

**Figure 3 materials-12-01100-f003:**
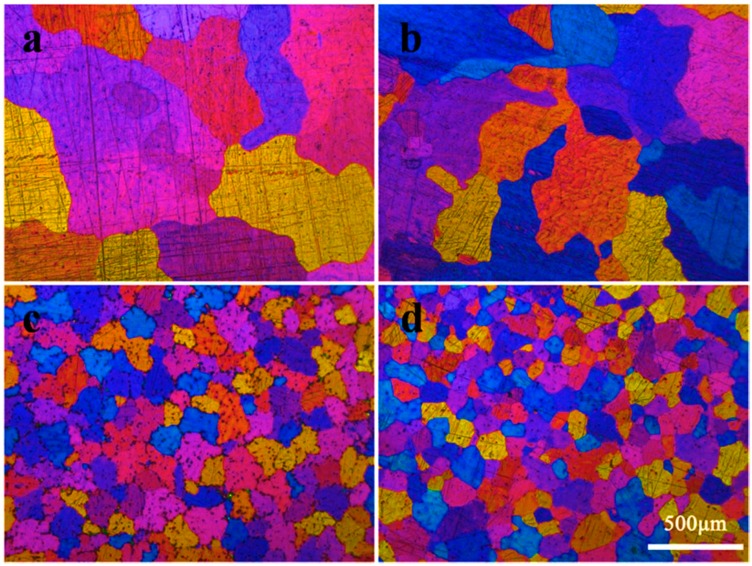
Optical microstructures of the AM50 alloys with (**a**) 0 wt. % B, (**b**) 0.05 wt. %B, (**c**) 0.1 wt. %B, (**d**) 0.15 wt. % B.3.3. Tensile Properties of *AM50* + *xB Alloys*.

**Figure 4 materials-12-01100-f004:**
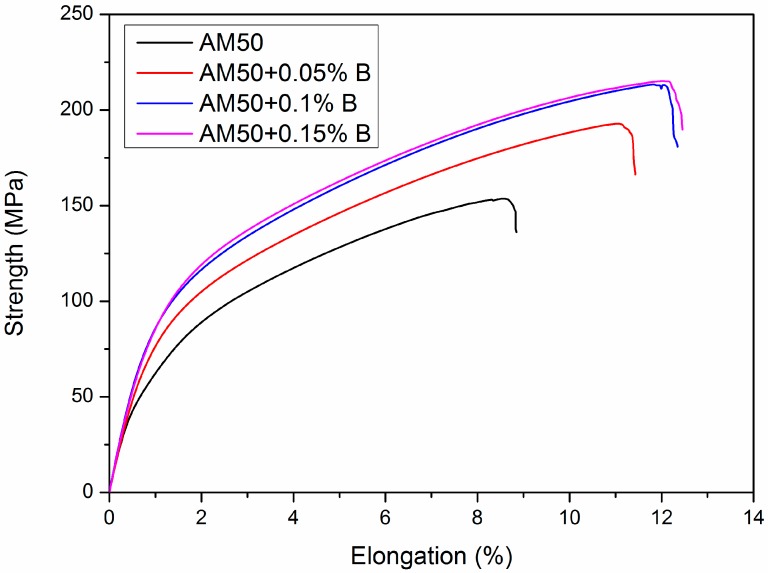
Stress-strain curves for the AM50+B specimens tested at room temperature.

**Figure 5 materials-12-01100-f005:**
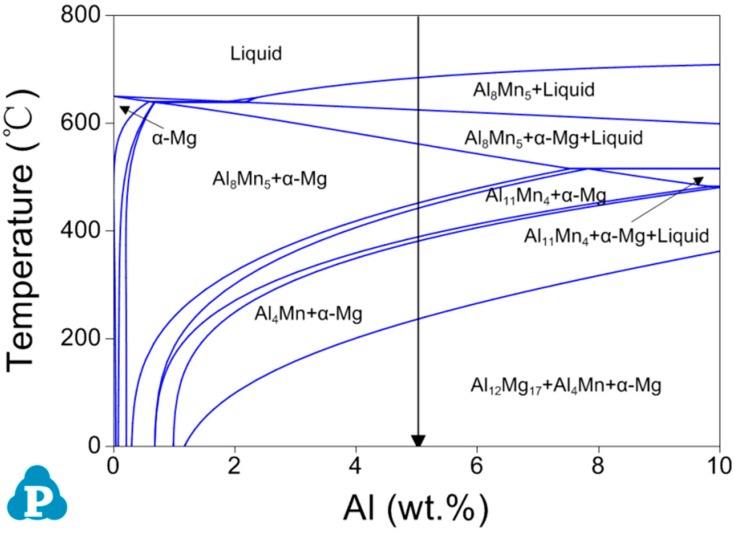
Section of Mg-0.5Mn-Al system calculated using the Pandat software.

**Figure 6 materials-12-01100-f006:**
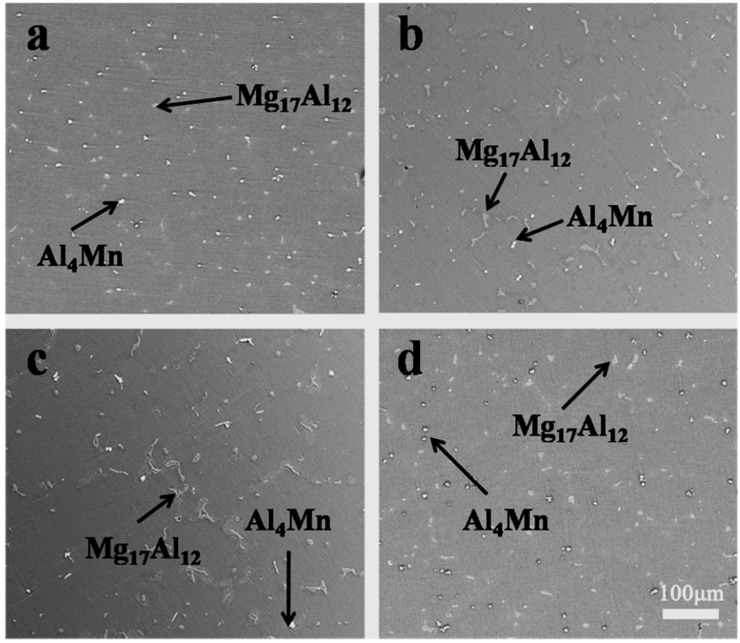
The SEM micrograph of the AM50 alloys with (**a**) 0 wt. % B, (**b**) 0.05 wt. %B, (**c**) 0.1 wt. %B, (**d**) 0.15 wt. % B.

**Figure 7 materials-12-01100-f007:**
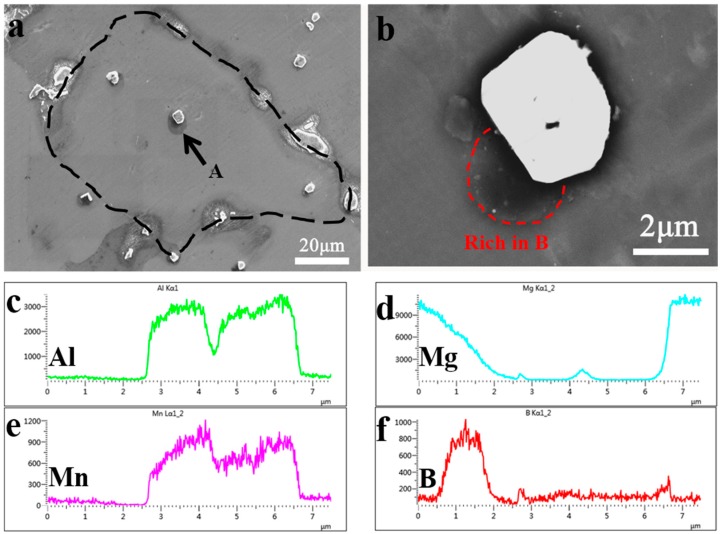
(**a**). SEM micrograph of 0.15wt. % B added AM50 alloy, (**b**). high magnification image of the particle A in BSE mode, (**c**–**f**) EPMA line scan results of Al, Mg, Mn and B, respectively.

**Figure 8 materials-12-01100-f008:**
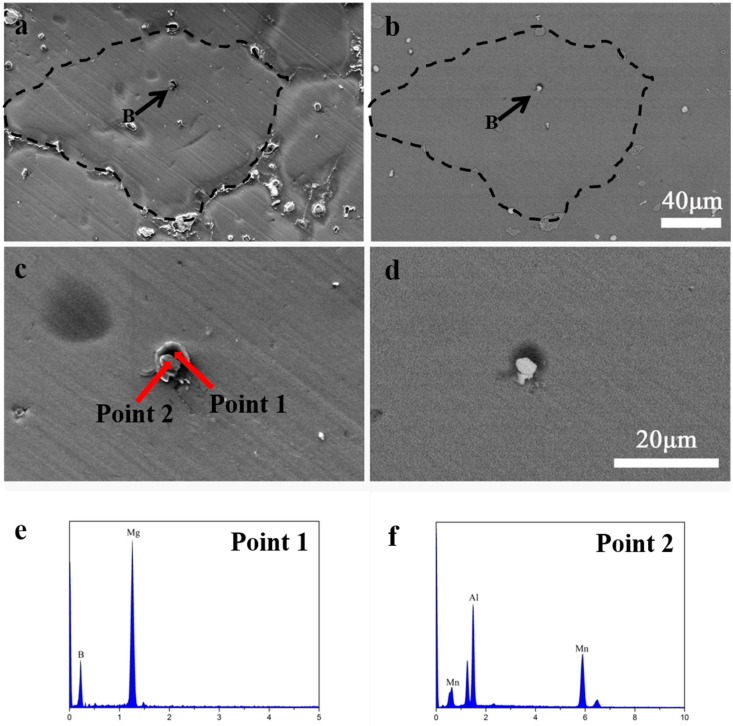
SEM micrograph of 0.1wt% B added AM50containing 0.1 wt. % B: (**a**). SE mode, (**b**). BSE mode (**c**). high magnification image of the particle B in SE mode (**d**). high magnification image of the particle B in BSE mode (**e**). the EPMA point result of point 1 (**f**). the EPMA point result of point 2.

**Figure 9 materials-12-01100-f009:**
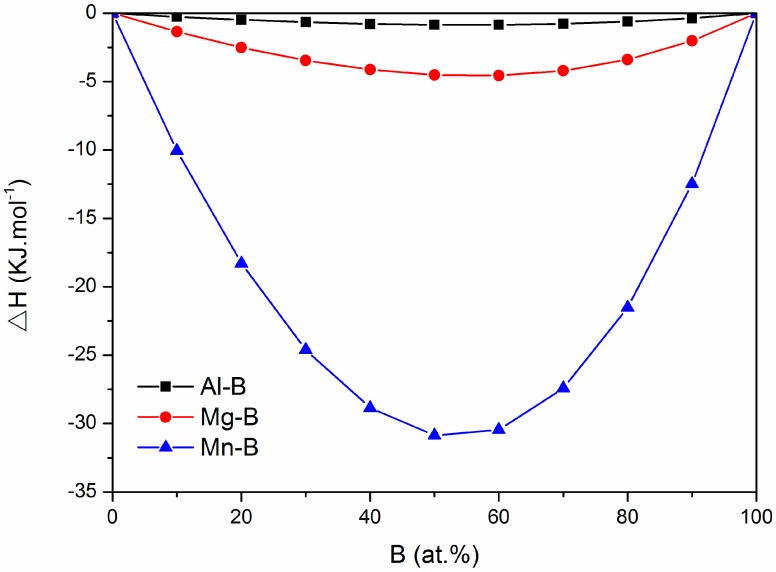
The enthalpies of Al-B, Mg-B and Mn-B compounds.

**Figure 10 materials-12-01100-f010:**
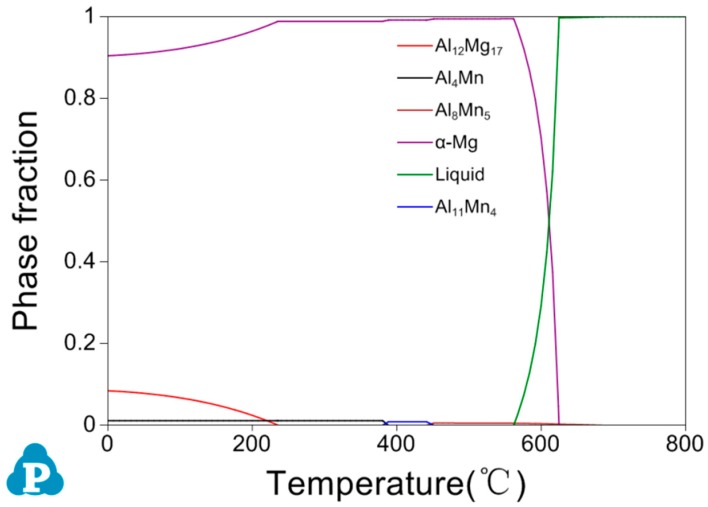
Phase formation of as-cast AM50.

**Table 1 materials-12-01100-t001:** Actual chemical compositions (wt. %) of the AM50 + xB (x = 0, 0.05, 0.1, 0.15 wt.%) alloys.

Sample	Al	Mn	B	Mg
AM50	4.6053	0.4582	0	Bal.
AM50 + 0.05B	4.8632	0.4763	0.0489	Bal.
AM50 + 0.10B	4.5891	0.4692	0.1162	Bal.
AM50 + 0.15B	4.7937	0.5131	0.1586	Bal.

**Table 2 materials-12-01100-t002:** The grain size and mechanical properties of AM50 + xB (x = 0, 0.05, 0.10, 0.15, wt.%).

	Grain Size (μm)	YTS (MPa)	UTS (MPa)	Elongation (%)
AM50	550 ± 150	48 ± 5	157 ± 4	8.3 ± 0.2
AM50 + 0.05%B	320 ± 96	70 ± 3	189 ± 5	10.3 ± 0.8
AM50 + 0.1%B	140 ± 21	87 ± 2	210 ± 7	12.1 ± 0.5
AM50 + 0.15%B	100 ± 16	94 ± 2	215 ± 6	12.3 ± 0.7

**Table 3 materials-12-01100-t003:** EPMA results of the Figure 6 (at. %).

Positions	Element				Possible Compounds
	Mg	Al	Mn	B	
Point 1	39	-	-	61	MgB_2_
Point 2	5	22	73	-	Al_4_Mn

**Table 4 materials-12-01100-t004:** Some Miedema model parameters.

Element	μ	$n_{ws}^{\frac{1}{3}}/d.u.$	*φ*/V	$V^{\frac{2}{3}}/{\mathrm{cm}}^{2}$
Al	0.10	1.39	4.20	4.6
Mg	0.07	1.17	3.45	5.8
Mn	0.04	1.61	4.45	3.8
B	0.07	1.55	4.75	2.8

**Table 5 materials-12-01100-t005:** The value of the parameter p in Miedema model.

Type	p
Mg-B	10.7
Al-B	10.7
Mn-B	12.35

**Table 6 materials-12-01100-t006:** Lattice parameters and misfit of AlB_2_, MgB_2_ and Mg.

	AlB_2_	Misfit on (0001)	MgB_2_	Misfit on (0001)	Mg
a (nm)	3.005	6.36	3.085	3.86	3.202
c (nm)	3.254		3.515		5.211

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
