# Peer review of "Effect of Boron on the Grain Refinement and Mechanical Properties of as-Cast Mg Alloy AM50"

_materials, 2019, doi:10.3390/ma12071100_

Round 1

Reviewer 1 Report

Zhang et al. report a case study on the effect of B addition on the grain refinement and mechanical properties of Mg-based alloys. The topic is relevant and the results are interesting. The obtained results have discussed in detail, using a wide broad spectrum of experimental techniques. Additionally, the authors show a solid knowledge of the state-of-the-art literature on the topic. The manuscript is well-structured and easy to follow. Thus the paper warrants the acceptance in Materials after these minor revisions:

-       All equations should be numbered along the manuscript.

Author Response

In the revised manuscript, the modification has been made and marked in red (see page 4-5).

Reviewer 2 Report

The manuscript addresses a interesting subject and is clearly presented. However some changes are necessary in order to publish it. Regarding the measurement of grain size, the authors should present in the procedure the number of counts did. With regard to the figures, some should be amended to make the results clearer. Referring to Figure 1, plane reflections should also be represented in the peaks. Regarding the EDS spectrum, the first peak is very intense. What does this peak correspond to? Figure 3 should present images without problems of metallographic preparation. Once the authors present OM images then they should present images of well-prepared samples. My suggestion is to replace these images with better quality ones.

Author Response

Comment 1: Regarding the measurement of grain size, the authors should present in the procedure the number of counts did. Response 1: For the grain size measurement, around 100 counts are recorded to calculate the average grain size. The corresponding modification is added and marked in Page 2, line 75. Comment 2: With regard to the figures, some should be amended to make the results clearer. Referring to Figure 1, plane reflections should also be represented in the peaks. Response 2: We have added the plane reflections to Fig. 1. Comment 3: Regarding the EDS spectrum, the first peak is very intense. What does this peak correspond to? Response 3: Regarding t this question, we have talked with the EDS operator. We believed the first intense peak does not correspond to any element since the voltage is close to 0 KV, and this peak comes from the facility. The similar peak is also found in Vlasceanu, M. et al’s (Vlasceanu, M., Sin, S. L., Elsayed, A., Ravindran, C. (2014). Effect of Al–5Ti–1B on grain refinement, dendrite coherency and porosity of AZ91E magnesium alloy. Cast Metals, 28(1), 39-46) work. The presence of this peak does not affect our results. Comment 4: Figure 3 should present images without problems of metallographic preparation. Once the authors present OM images then they should present images of well-prepared samples. My suggestion is to replace these images with better quality ones. Response 4: We admit that there are still some visible scratches in Fig. 3 a and Fig. 3 d. However, both pictures can present the grain morphology information clearly. As magnesium alloy is very soft and the scratches are not easy to be removed and we do not have other better pictures now, we would like to keep Fig. 3 a and Fig. 3 d as the present form.

Reviewer 3 Report

This manuscript has studied the effect of the B to the grain structure and mechanical properties of the Mg-based alloy. The data clarified the mechanisms of B refinement. The topic is interest and the result can improve the properties of both cast and wrought Mg based alloys, because they have a hard-deformable HCP lattice and their ductility directly affected by grain size.

Before paper will be published several can be taken into account by authors to  improve results clarity

In the abstract authors presented the mechanical properties only one B-containing alloy. It is recommended to improve the abstract by specify the B effect to the mechanical properties.

Carbon inoculation is only one method of the grain refinement of Mg-Al based alloys. The introduction part can be improved by full describing the refinement phenomena and grain refinement methods.

It is recommended to use “Al-bearing” instead “Al bearing” as authors used “B-containing” etc

124 line a-Mg correct to α-Mg

145 line  “the amount of the B containing compound increased  with increasing with B content” I  think it is trivial, please rephrase

Furthermore, the conclusion section of your work does not present any conclusions but is rather a short list of facts reported in earlier portions of the paper. Please note, that a good concluding section notes the limitations of the study. It should mention the scope for further research as well as the implications/application of the study.

Author Response

Comment 1: In the abstract authors presented the mechanical properties only one B-containing alloy. It is recommended to improve the abstract by specify the B effect to the mechanical properties. Response 1: The following sentence is added in the abstract to specify the effect of B on the mechanical properties “The addition of B to as cast AM50 alloy results in improved mechanical properties of AM50+xB alloys. For instance, the”, as marked in line 18, 19. Comment 2: Carbon inoculation is only one method of the grain refinement of Mg-Al based alloys. The introduction part can be improved by full describing the refinement phenomena and grain refinement methods. Response 2: We have added the following sentences “For instance, Zhang et al refined the grains of AZ91 with melt conditioned high pressure die casting [10], Jiang et al. used Al2Y as a grain refiner for Mg-Y-Al alloys [12].”, as seen in line 35-36. Besides, the role of carbon has many similarities with boron, such as the formation of a heterogeneous nucleation core and its low price, so it is described more in detail in the introduction. Comment 3: It is recommended to use “Al-bearing” instead “Al bearing” as authors used “B-containing” etc Response 3: In the revised manuscript, the “Al-bearing” is used to replace “Al bearing” (see page 1, line 33). Comment 4: 124 line a-Mg correct to α-Mg Response 4: In the revised manuscript, the “a-Mg” is modified to “α-Mg” (see line 129). Comment 5: 145 line “the amount of the B containing compound increased with increasing with B content” I think it is trivial, please rephrase Response 5: In the revised manuscript, “the amount of the B containing compound increased with increasing with B content” has been changed to “the amount of the B containing compound (either Mg-B or Al-B) increased with increasing the B content”, as marked red in line 151-152 Comment 6: Furthermore, the conclusion section of your work does not present any conclusions but is rather a short list of facts reported in earlier portions of the paper. Please note, that a good concluding section notes the limitations of the study. It should mention the scope for further research as well as the implications/application of the study. Response 6: We fully agree with that a good conclusion should include the limitations and outlook of the further research. Thus, the following sentences are added “which is very important for the die cast AM50 magnesium alloy parts”; “However, the MgB2 particle is not yet identified by TEM in this study, which will be further investigated in the future.”

Reviewer 4 Report

Manuscript ID materials-470316  “Effect of boron on the grain refinement and mechanical properties of as-cast Mg alloy AM50” I recommend publishing this paper after major revisions according by following suggestions.

The main comments:

Please, fill and correct your article according to several suggestions

The results of tensile test - add stress-strain curve, and also correct the results;  true stress vs true strain - correct and justify

The value of tensile yield stress determine and compare

Add instrumental method (operating parameters, QA/QC procedure) and detailed descript methods

Improve the figures (quality of pictures, and description in the picture

line 162, 168, etc. , p. 4 – add numbering of equations

line 205,206, p. 5 – improve and divide the phase transformation and solidification process

Chapter References – over 77% references are older than 5 years, is only 23% of used scientific citations are date from 2014, others are older than 5 years; please add the other new references no older 5 years. And also correct your list of references according to guidelines for authors.

I recommend accept manuscript after major revision.  

Sincerely

Author Response

Comment 1: The results of tensile test - add stress-strain curve, and also correct the results; true stress vs true strain - correct and justify. The value of tensile yield stress determine and compare Response 1: AM50+XB alloy does not show obvious necking under tension, so the mechanical properties of AM50+XB are presented as nominal stress-strain curve rather than true stress-strain curve. This is presented in many related reports as well, so we think it is more reasonable to use nominal stress-strain here. Comment 2: Add instrumental method (operating parameters, QA/QC procedure) and detailed descript methods Response 2: We tried our best but we did not fully understand the reviewer’s comment, we think our description on all the experimental procedures is in detail. Please specify which method should we add parameters. Comment 3: Improve the figures (quality of pictures, and description in the picture Response 3: We have improved the contrast of Fig.3 and Fig. 6. Comment 4: line 162, 168, etc. , p. 4 – add numbering of equations Response 4: In the revised manuscript, the modification has been made and marked in red (see page 4-5). Comment 5: line 205,206, p. 5 – improve and divide the phase transformation and solidification process Response 5: In the revised manuscript, the modification has been made and marked in red (see line 213-214) Comment 6: Chapter References – over 77% references are older than 5 years, is only 23% of used scientific citations are date from 2014, others are older than 5 years; please add the other new references no older 5 years. And also correct your list of references according to guidelines for authors. Response 6: We have corrected the format of references and added 6 references later than 2014, as shown below: [7] Nagasivamuni, B.; Wang, G.; StJohn D.H.; Dargusch, M.S. Effect of ultrasonic treatment on the alloying and grain refinement efficiency of a Mg - Zr master alloy added to magnesium at hypo- and hyper-peritectic compositions, Journal of Crystal Growth, 2019, 512, 20-32. [10] Qiu, W.; Liu, Z.Q.; Yu, R.Z.; Chen, J.; Ren, Y.J.; He, J.J.; Li, W.; Li, C. Utilization of VN particles for grain refinement and mechanical properties of AZ31 magnesium alloy. Journal of Alloys and Compounds, 2019, 781, 1150-1158. [11] Zhang, Y.; Patel, J. B.; Wang, Y.; Fan, Z. Variation improvement of mechanical properties of Mg-9Al-1Zn alloy with melt conditioned high pressure die casting. Materials Characterization, 2018, 144, 498-504. [12] Jiang, Z.T.; Feng, J.; Chen, Q.W.; Jiang, S.; Dai, J.H.; Jiang, B.; Pan, F.S. Preparation and Characterization of Magnesium Alloy Containing Al2Y Particles. Materials, 2018, 11(9), 1748. [14] Du, J.; Yao, Z.; Han, S.; Li, W. Discussion on grain refining mechanism of AM30 alloy inoculated by MgCO3. Journal of Magnesium and Alloys. 2017, 5, 181-188. [17] Koltygin, A.; Bazhenov, V.; Mahmadiyorov, U. Influence of Al–5Ti–1B master alloy addition on the grain size of AZ91 alloy. Journal of Magnesium and Alloys. 2017, 5, 313-319.

Round 2

Reviewer 1 Report

The authors have improved their manuscript and replied to the referees' comments accordingly.

Reviewer 2 Report

The paper can be now accept. 

Reviewer 3 Report

This manuscript can be published in Materials, I have no additional significant suggestions. Please, carefully read paper in a proof process, I think the style of the last sentence can be improved.